# A Pot-Like Vibrational Microfluidic Rotational Motor

**DOI:** 10.3390/mi12020177

**Published:** 2021-02-11

**Authors:** Suzana Uran, Matjaž Malok, Božidar Bratina, Riko Šafarič

**Affiliations:** Faculty of Electrical Engineering and Computer Science, University of Maribor, Koroška c. 46, 2000 Maribor, Slovenia; suzana.uran@um.si (S.U.); matjaz.malok@student.um.si (M.M.); riko.safaric@um.si (R.Š.)

**Keywords:** microfluidics, micro-sized streaming of a liquid in the pot-like structure, micro-sized rotational motor, piezoelectric driven vibrations

## Abstract

Constructing a micro-sized microfluidic motor always involves the problem of how to transfer the mechanical energy out of the motor. The paper presents several experiments with pot-like microfluidic rotational motor structures driven by two perpendicular sine and cosine vibrations with amplitudes around 10 μm in the frequency region from 200 Hz to 500 Hz. The extensive theoretical research based on the mathematical model of the liquid streaming in a pot-like structure was the base for the successful real-life laboratory application of a microfluidic rotational motor. The final microfluidic motor structure allowed transferring the rotational mechanical energy out of the motor with a central axis. The main practical challenge of the research was to find the proper balance between the torque, due to friction in the bearings and the motor’s maximal torque. The presented motor, with sizes 1 mm by 0.6 mm, reached the maximal rotational speed in both directions between −15 rad/s to +14 rad/s, with the estimated maximal torque of 0.1 pNm. The measured frequency characteristics of vibration amplitudes and phase angle between the directions of both vibrational amplitudes and rotational speed of the motor rotor against frequency of vibrations, allowed us to understand how to build the pot-like microfluidic rotational motor.

## 1. Introduction

The miniaturized microfluidic rotational motors have been at the forefront of the research efforts of scientists and engineers who have been developing micro rotational motors as a part of microelectromechanical systems (MEMS) for more than three decades.

The first microfluidic rotational motors, published in the 1990s [1], had rotary gear trains made on the silicon rotor (diameter 60 µm to 1600 µm). They were built into micro-sized channels, where fluidic linear stream of liquids was used as the driving forces for the gears. The maximal rotational speed was 390 rad/s with a high max torque of 8.7 pNm. Their rotational speed was controlled in only one direction. The rotary gear trains with the rotor were closed in a micro-sized channel, and the rotational mechanical power could not be transferred outside the micro-sized channel.

The mm-sized rotor of a motor, controlled in both directions by electro wetting, was reported in [2]. The electro-wetting microfluidic motor was made by a 3.0 μL liquid droplet between two electric plates. The maximal rotational speed 19 rad/s was performed on a rotor with a diameter of 2 mm. There was no report of max torque. The rotational mechanical power could not be transferred outside the motor.

Amjadi et al. [3] published rotating the thin film of liquid between the electric plates using an electric field. The experiment had not reported the actuation of the disc on the surface of the liquid, probably because the layer of liquid was too thin.

Shilton et al. and Yeo et al. [4,5,6] reported succeeding rotating the disc on the thin layer of liquid. Rotational microfluidic motors with micrometer to millimeter-sized diameters of rotors, used the vibration energy of surface acoustic waves (SAW) at the frequency 20 MHz to drive a rotor floating in the thin layer of liquid. This method used two oppositely directed surface acoustic waves to drive a fluid-coupled disc with a diameter of 5 mm. The disc was controlled in both directions at rotational velocities of 235 rad/s and with maximal torque of 60 nNm. The rotational mechanical power was not able to be transferred outside the motor.

Specially designed rotors (diameter 600 µm) with 2–6 sharp edges used one source of vibration at the frequency 4.4 kHz with amplitudes up to 14 µm [7]. The sharp edges produced the whirls in the liquid, which were able to rotate the rotor with maximal rotational speed of 125 rad/s. The rotational velocity was controlled in only one direction. The maximal torque was not reported. Again, the rotational mechanical power was not able to be transferred outside the motor. A similar approach from Feng et al. [8] of streaming the whirls nearby the sharp edges of structures, induced by vibrations, was used to rotate the living cells.

Li et al. [9] demonstrated how to use the Janus self-propelled micromachine for rotation of an aluminum sphere with diameter of 100 µm in an NaOH solution, with a life span less than 200 s. There was no report about maximum torque, and the method was not suitable to transfer rotation mechanical power. 

Wang et al. and Kao et al. [10,11] demonstrated how a microfluidic flow, created by the interaction between a bubble and acoustic waves at 100 kHz, could be exploited to rotate a rotor at a speed of 65 rad/s. The winged seed of a maple tree inspired the rotor shape, with diameter 65 µm. There was no report about the maximum torque.

In the experiments of Hayakawa et al., Zhang et al., and Tan et al. [12,13,14,15], the circular vibrations produced the whirls in the liquid around a pillar with 10–100 µm diameter. The whirls between three pillars, or SAW in the liquid, were used to rotate micro-sized living cells slowly. The highspeed rotation of a 10 µm object around one pillar was presented in [16]. 

The same method as in the previous cited reference was used in the research of Uran et al. [17], where whirling of the liquid around the pillar in a droplet of water was used to rotate a disc (diameter 350 µm) centered with the pillar. They reported the maximum rotation speed around 26 rad/s, controllable in both directions and with maximal torque of 0.2 pNm. Again, the rotational mechanical power was not able to be transferred outside the motor.

The papers from Hayakawa et al. and Uran et al. [12,13,17] with their mathematical model of whirling liquid around the central pillar were partly the source of an idea used for the design of our mathematical model of a pot-like microfluidic rotational motor. All of the above mentioned vibrational micromotors, except [7], including the rotating of whirls of a thin layer of water in the electric field and rotating of micro-sized living cells [1,2,3,4,5,6,8,9,10,11,12,13,14,15,16], used the design of the rotational disk (rotor), or rotated living cell, which was not fixed in the space of a motor with an axis. This means that these approaches are not able to drive, for example, the gear-boxes, or, in other words, they were not able to transfer rotational mechanical power out of the motor. An overview of above discussed micromotors is presented in Table 1.

The review paper of Ahmed et al. [18], with the microfluidic rotational streaming of liquid presented the research activities of the topics until 2018 with 227 references. The authors of the review paper divided the area of microfluidic rotation motors to mechanisms employed to generate centrifugal flows by: Passive actuation mechanisms (actuation was caused by geometrical and topological properties), and active actuation mechanisms (caused by electro kinetic, acoustic, thermal, mechanical, and pneumatic actuation). Our paper presents the novel application of a microfluidic rotational motor using the acoustic (vibrational) actuation, so it was compared predominantly to the similar approaches from the area of microfluidic rotational motors with acoustic actuation [4,5,6,7,8,9,10,11,12,13,14,15,16], two examples from the subcategory of electro kinetic actuation [2,3], and one example from the category of passive actuation mechanisms [1], due to historical reasons. Altogether the compared applications are only a small part of the microfluidic rotational motors presented in [18]. Another, more focused review from Jalal et al. [19], presented the theory of microstreaming and microfluidic applications induced by vibrations and the influence of different geometrical sources of streaming (sphere, wall, pillar …).

The focus of our paper is to solve the obvious drawback of almost all cited microfluidic motors, which are unable to transfer the rotational power out of the motor. The paper presents the novel and unique pot-like microfluidic vibrational driven rotational micromotor. The motor produces the controllable rotation of the disk (rotor) attached rigidly on the central axis, which was fixed in the motor with upper and lower bearings, and, therefore, able to transfer the rotational mechanical power out of the motor. The controllability was achieved by the change of frequency inside the frequency region 200–550 Hz, by change of phase angle between sine and cosine vibrations and by change of vibrations’ amplitudes from 1–10 µm. The paper presents the construction and control of a pot-like vibrational microfluidic motor actuated by the circular vibrations. A circular water stream is created inside the pot-like structure of the motor, which drives the rotor (diameter of 350–400 µm) and rigidly attached axis to the rotor of a microfluidic motor. The rotational velocity of the motor axis is controlled in both rotational directions from −15 rad/s to +14 rad/s. The maximal torque is estimated at 0.1 pNm with practically no wear. The motor is sealed in the pot-like structure, which prevents the evaporation of the liquid (water) from the motor, which prolongs the running of the motor without maintenance (replacing of evaporated water) to about 9–10 h. The presented microfluidic rotational motor can be used in MEMS applications, where a reliable submillimeter size rotational motor is needed.

## 2. Materials and Methods

### 2.1. Materials

The circular wall (a pot-like structure) with inner diameter 500 μm to 2000 μm were made from a plastic tube, purchased from the Micro+Polo d.o.o. company, Maribor, Slovenia. The vibrational pillars were made from nickel wire with the diameter 50 µm, purchased from Alfa Aeser GmbH & Co. KG, Kandel, Germany, or from optic fiber with the diameter 80 µm without the plastic envelope, which was removed from the fiber. The micro-sized rotor was made from a piece of a flattened Styrofoam ball, the glass supporting plate (7 by 7 by 0.2 mm) was made from microscopic glass, purchased from the Micro+Polo d.o.o. company, Maribor, Slovenia. The piezoelectric stack actuators, MPO Piezo Stacks MPO-050015 with electric capacitance at 1 V RMS, 1 kHz, *C_p_* = 1.1 µF, resonant frequency of unloaded piezo actuator *f_res_* = 105 kHz, maximal push force *F_PUSH_* = 1500 N, maximal pull force *F_PULL_*
_=_ 50 N, piezoelectric large-signal deformation coefficient *d*_33_ =10^−8^ m/V, number of stages *n* = 10, and stiffness *k* = 75 N/µm, were purchased from the company Nanofaktur, Villingen-Schwenningen, Germany. The piezoelectric actuators were used for generating of cosine and sine mechanical vibrations. The transparent ultraviolet glue Bondic was purchased from the company VIKO UG, Munich, Germany, and was used to join the parts of the microfluidic motor together. A high voltage amplifier EK29 KIT (Power Booster Evaluation Kit) with a high voltage operational amplifier was produced by the company Apex Microtechnology, Tucson, Arizona, USA. The vibration mechanism had a metallic base that was attached rigidly to the optical microscope (EUROMEX, Arnhem, the Netherland), and its tip was capable of being moved into the focus of the microscope lens.

### 2.2. Laboratory Set-Up

The laboratory set-up was made by a vibrational mechanism and a supporting electronic device. The vibrational mechanism is presented in Figure 1a, and its scheme in Figure 1c. PLA yellow plastic, made by a 3D printer was used as a supporting frame, was attached on the mechanical stage of an optical microscope and was moving in the x-y directions. The L-shaped vibrational device was made by two perpendicularly mounted piezo actuators via an aluminum L-shaped connection part. The vibrational device was attached via its metallic base to the yellow frame. The glass supporting plate was attached on the tip of the L-shaped vibrational device via an aluminum L-shaped connection part. All parts, the piezo actuators, the aluminum connection parts, the metallic base and the glass plate, were glued to each other with the UV glue. The microfluidic rotational motor was mounted in the center of the supported glass plate. The microfluidic rotational motor on the glass supported plate was focused into a suitable position for the experiment observation between the microscope lenses and the light source in the bottom of the microscope.

The piezo electric actuators supplied by cosine and sine AC voltage presented in Figure 1b (maximal peak to peak voltage *V_pp_* = 190 V). The piezo actuator produced the vibrations in both x and y directions (see Figure 1c), which was transferred as vibration amplitudes *a*_1_ and *a*_2_ to the pot-like micro motor installed in the center of the glass plate. The cosine and sine vibrations *a*_1_ and *a*_2_, induced by the adjustable AC voltages, produced the circular vibrational movement with an adjustable diameter between 0 and 13 μm of the glass plate (see Figure 1d). The circular vibration of the pot-like mechanism on the glass plate produced the circular streaming of the liquid (water) in the pot, which was used as a driving force to rotate the disk floating on the water’s surface in the pot. The different pot-like mechanisms are described exactly in Section 2.3.4.

The electronic part of the laboratory set-up is presented in Figure 2. A personal computer was used as the man-machine communication and was connected via USB to a UART cable to a dsPIC microcontroller. The dsPIC microcontroller calculated two AC output signals (0 V to 3 V) for the *X* axis and *Y* axis piezo actuators with changeable frequency (1 Hz to 10 kHz), the changeable phase between the *X*-axis and *Y*-axis output voltages (−180° to + 180°) and changeable amplitudes of both voltages (-20 V to +170 V). The generated sine (for the *X*-axis) and cosine (for the *Y*-axis) signals were amplified with two EK29 KITs amplifiers to maximal amplitudes +170 V and −20 V and maximal effective current *I_eff, max_* = 1.5 A. The amplified voltage for both axes were used to supply both piezo actuators in the mechanical part of the laboratory set-up (see Figure 1). The core of the EK29 KIT amplifier is a high voltage operational PB51amplifier produced by APEX Microtechnology (Tucson, Arizona, USA).

### 2.3. Methods

#### 2.3.1. The Mathematical Model of the Microfluidic Streaming of Water around the Circular Vibrating Pillar

The mathematical model of the vibration-induced whirling flow of incompressible fluid around the long infinitive circular vibrating pillar with micro-sized diameter and its development, were presented in [12,13,17]. The derivation of the mathematical model follows the approach in Holtsmark [20], where the steady term $\psi_{st}^{\left(1\right)}$ was calculated for linear vibration. The same approach was used in Hayakawa et al. [12], but the circular vibration was used for the calculation of steady term $\psi_{st}^{\left(1\right)}$. We followed the development of a solution of stream function based differential equations for the motion of incompressible viscous fluid in a two-dimensional space under assumption that the fluid obeys the applied circular vibration at an infinite distance from the micro pillar, and that the fluid speed is zero at the surface of the micro pillar in [12]. Therefore, the steady term of the solution for the stream function $\psi_{st}^{\left(1\right)}$ was calculated by Equations (1)–(8):(1)$$\psi_{st}^{\left(1\right)}\;={\;r}^{4}\left(\frac{1}{48}\int_{a}^{r}\frac{1}{x}\rho \left(x\right)\mathrm{dx}+c_{1}\right)+{\;r}^{2}\left(-\frac{1}{16}\int_{a}^{r}x\rho \left(x\right)\mathrm{dx}+c_{2}\right)+\;\;\left(\frac{1}{16}\int_{a}^{r}x^{3}\rho \left(x\right)\mathrm{dx}+c_{3}\right)\;+\;\frac{1}{r^{2}}\left(-\frac{1}{48}\int_{a}^{r}x^{5}\rho \left(x\right)\mathrm{dx}+c_{4}\right)$$
where
(2)$$\rho \left(x\right)=\frac{2\pi^{3}f^{3}A^{2}}{\eta^{2}}\left[2Y+2Y^{*}-2\frac{a^{2}}{r^{2}}\mathrm{CD}-2\frac{a^{2}}{r^{2}}C^{*}Z-4{\mathrm{YY}}^{*}+4{\mathrm{ZZ}}^{*}\right],$$
(3)$$c_{1}=-\frac{1}{48}\int_{a}^{\infty }\frac{1}{x}\rho \left(x\right)\mathrm{dx},$$
(4)$$c_{2}=\frac{1}{16}\int_{a}^{\infty }x\rho \left(x\right)\mathrm{dx},$$
(5)$$c_{3}=\frac{a^{4}}{16}\int_{a}^{\infty }\frac{1}{x}\rho \left(x\right)\mathrm{dx}-\frac{a^{2}}{8}\int_{a}^{\infty }x\rho \left(x\right)\mathrm{dx},$$
(6)$$c_{4}=-\frac{a^{6}}{24}\int_{a}^{\infty }\frac{1}{x}\rho \left(x\right)\mathrm{dx}+\frac{a^{4}}{16}\int_{a}^{\infty }x\rho \left(x\right)\mathrm{dx}.$$

The functions *Y*, *Z*, *C,* and *D* were calculated from the Hankel functions of the first kind $H_{n}^{\left(1\right)}$ and from Hankel functions of the second kind $H_{n}^{\left(2\right)}$ with the next equations: (7)$$Y=\frac{H_{0}^{\left(1\right)}\left(\varepsilon .x\right)}{H_{0}^{\left(1\right)}\left(\varepsilon .a\right)},\;Z=\frac{H_{2}^{\left(1\right)}\left(\varepsilon .x\right)}{H_{0}^{\left(1\right)}\left(\varepsilon .a\right)},\;C=\frac{H_{2}^{\left(1\right)}\left(\varepsilon .a\right)}{H_{0}^{\left(1\right)}\left(\varepsilon .a\right)}\;,\;D=\;\frac{H_{2}^{\left(2\right)}\left(\varepsilon z.x\right)}{H_{0}^{\left(1\right)}\left(\varepsilon .a\right)},$$
(8)$$\varepsilon ={\left(i\frac{2\pi f}{\eta }\right)}^{\frac{1}{2}}\;,\;\varepsilon z={\left(-i\frac{2\pi f}{\eta }\right)}^{\frac{1}{2}}.$$

The parameters in Equations (1)–(8) are: *f* is the frequency and *A* is the amplitude of the applied circular vibration, *r* is the distance from the pillar, η is the kinematic viscosity of the fluid, and *a* is the radius of the pillar.

The velocity vector components in polar coordinates of the steady fluid whirling flow were calculated from Equations (1) using the next equations:(9)$$v_{\theta }^{p}=v_{t}=\frac{{\delta \psi }_{\mathrm{st}}^{\left(1\right)}}{\delta r}\;,{\;v}_{\mathrm{ra}}^{p}=\frac{1}{r}\frac{{\delta \psi }_{\mathrm{st}}^{\left(1\right)}}{\delta \theta }=0.$$

The $v_{ra}^{p}$ in Equations (9) is the radial component of the induced flow velocity in the polar coordinates (the prefix p means polar) and is equal to zero, because the steady term of the stream function $\psi_{st}^{\left(1\right)}$ does not depend on the polar variable *θ*. The $v_{\theta }^{p}$ component is perpendicular to the $v_{ra}^{p}$ velocity component, and is tangential to the rotation of liquid around the micro pillar, so we called it tangential velocity $v_{t}$. The length and Cartesian components of the tangential velocity vector $\vec{v_{t}}$ in the Cartesian coordinates in a clockwise direction (CW) are given in Equation (10):(10)$$\vec{v_{t}}=\left[-v_{t}\cdot \sin\left(\theta \right),\;-v_{t}\cdot \cos\left(\theta \right)\right].$$

Based on the liquid tangential velocity vector a rotational velocity vector was defined. Both the liquid tangential velocity $v_{t}$ [m/s] and rotational velocity $v_{r}$ [rad/s] (see Equation (11)) of the liquid stream around the micro pillar were calculated using numerical derivation in MATLAB.
(11)$$v_{r}=\frac{v_{t}}{r}=\frac{1}{r}\frac{{\delta \psi }_{\mathrm{st}}^{\left(1\right)}}{\delta r},\;\vec{v_{r}}\;=\frac{\vec{v_{t}}}{r}\;.$$

The calculated rotational speed $v_{r}$ [rad/s] of the rotated water stream around the circular vibrated pillar against the distance from the pillar *r* are shown in Figure 3a, and its amplitude in Figure 3b. The simulated tangential speed and its amplitude around the same pillar are presented in Figure 4a,b, respectively. The liquid was water with kinematic viscosity η=1.0 × 10^−6^ m^2^ s^−1^, the radius of the pillar was *a* = 40 µm, the amplitude and frequency of the circular vibration were *A* = 8 µm and *f* = 500 Hz, respectively. Equations (1)–(11) showed that the rotational and tangential velocities can be increased by increased amplitude *A*, frequency *f*, and decreased fluid’s kinematic viscosity η. 

#### 2.3.2. The Mathematical Model of Microfluidic Streaming of Water around Several Pillars

More complex layouts of the pillars were studied, and the streaming of the liquid around and between them. Every single pillar with the same circular vibration affects the streaming of liquid around all the involved pillars. Equation (12) shows the contribution of all circular vibrating pillars to the liquid streaming’s rotational and tangential vectors $\vec{v_{r}}$ and $\vec{v_{t}}$ for every single position in a 2D space (x-y plane) for $-X_{o}<x<+X_{o}$ and $-Y_{o}<y<+Y_{o}$, where $\mp X_{o}$ and $\mp Y_{o}$ are the upper and lower limits of the *x* and *y* coordinates in the x-y plane.
(12)$$\vec{v_{r}}\;=\sum_{i=1}^{n}\vec{v_{ir}}\;,\;\vec{v_{t}}\;=\sum_{i=1}^{n}\vec{v_{it}}$$
where *n* is the number of pillars in the complex layout. Such a layout with only four pillars to present rotational velocities is shown in Figure 5.

An even more complex layout of 15 pillars which would be suitable for streaming of water inside an open pot-like layout of the pillars’ configuration is presented in Figure 6. The inner diameter of the pot-like shape of pillars is *D* = 500 µm.

If the pillars overlapped physically, like in Figure 7, Figure 8 and Figure 9, then the mathematical Equation (12) was modified so as not to take into account the water streaming caused by the parts of the pillars which are outside the pot, or part of the wall.

#### 2.3.3. Mathematical Model of Microfluidic Streaming of Water in the Closed Pot-Like Configurations of Pillars

Our goal was to construct the closed pot-like configuration, where circular configured pillars from the side and plates on the bottom and the top prevented the evaporation of the liquid (water) from the pot. We tested different pot-like configurations of pillars to estimate the rotational velocity of the water stream inside the pot. The most suitable configurations for the pot-like microfluidic motor were found and presented in Figure 7, Figure 8 and Figure 9, where the rotational velocity of water streaming was presented only inside the pot. 

Figure 7 presented the simulation results of the model of completely closed configurations of pillars, where 24 pillars even overlap each other. The maximal rotational velocity was approximately 9 rad/s at a distance 50 µm away from the pot wall, and then the rotational velocity was decreased exponentially to zero in the middle of the pot. The inner diameter of the pot was *D* = 500 µm, and the amplitude and the frequency of the circular vibration were *A* = 8 µm and *f* = 500 Hz, respectively. 

Figure 8 presented the same configuration of pillars as the previous Figure 7 with one exception, a central pillar with a radius *acp* = 40 µm was added in the middle of the pot. Here the average rotational speed of the water stream was increased dramatically in comparison with the previous Figure 7. The highest rotational velocity was approximately 14 rad/s about 50 µm away from the central pillar, the lowest rotational velocity was approximately 5 rad/s about 150 µm away from the central pillar, and the rotational velocity nearby the pot wall was again increased to 9 rad/s.

It was also possible to increase the average simulated rotational velocity of water streaming inside the pot in two ways (see Figure 9). 

The first one in Figure 9, where the radius of the central pillar was increased to *acp* = 120 µm at the fixed inner diameter of the pot *D* = 500 µm, and the second one, where the gap between the pillar and the pot wall was decreased to approximately 150 µm, with simultaneously adjusting the radius of the central pillar and/or the inner diameter of the pot.

The Matlab code of the mathematical model is attached as Appendix A.

#### 2.3.4. Presentation of the Pot-Like Microfluidic Rotational Vibrational Motor Types

After intensive tests with the mathematical model, we decided to perform several laboratory experiments with 7 types of pot-like microfluidic vibrational rotational motors whose basic mathematical models were presented by Figure 7, Figure 8 and Figure 9.

For the first type of motor presented, only a vibrational pot-like mechanism was made of 1–2 mm high transparent plastic tube attached rigidly to the glass support plate with UV glue. It was used for observation of the streaming of the water and its rotational speed inside the pot with inner diameter of approx. 500 µm. The cross-section of the motor construction is presented in Figure 10a.

The second type of motor was called a pot-like microfluidic rotational vibrational motor with free-floating rotating disc. The rotating disc was floating and rotating freely in the convex shape of the water surface inside a pot with the inner diameter of approx. 500 µm. The diameter of the disc was approximately 150 µm. The disc was made from a Styrofoam ball. The motor used the principle presented in Figure 7. The cross-section of the motor construction is presented in Figure 10b.

The third type of motor, called a pot-like microfluidic rotational vibrational motor with concave shape of the water surface, had a floating disc on the top of the concave shape of the water surface. The disc had the axis attached in the center of disc. The axis was used to fix the disc through the elongated (1 mm long) sliding bearing, made from a glass tube with 100 µm inner diameter. The bearing was attached in the middle of a transparent foil (width approx. 250 µm) with UV glue. The transparent foil was attached on the three or more pillars connected directly to the glass supporting plate. The motor used the principle presented in Figure 7. The cross-section of the motor construction is presented in Figure 10c.

The fourth type of motor, called a pot-like microfluidic rotational vibrational motor with centrally mounted floating disc on the central pillar with radius 40 µm, had a central pillar made by optical fiber without the plastic envelope, which was attached by the UV glue in the center of the pot. The more or less circular shape of the disc with the diameter approximately 350 µm had a hole through it in the center of the floating disc. The motor used the principle presented in Figure 8. The cross-section of the motor construction is presented in Figure 10d.

The fifth type of the tested motor, called a pot-like microfluidic rotational vibrational motor with centrally mounted floating disc on the central pillar with radius 40 µm and attached outer rotating axis, used the outer rotating axis, which rotated around the central pillar and transferred the rotation movement of the disc out of the pot. It was also closed from above with transparent plastic foil, which allowed observation of the rotation of the disc inside the pot, and, at the same time, prevented the evaporation of the liquid out of the pot. The motor used the principle presented in Figure 8. The cross-section of the motor construction is presented in Figure 10e.

The sixth type of the motor, called a pot-like microfluidic rotational vibrational motor with centrally mounted floating disc on the central pillar with radius 120 µm and attached outer rotating axis, was similar to the previously explained type of motor, the only difference being that the central pillar’s radius was increased to 120 µm. This type of motor used the principle presented in Figure 9. The cross-section of this type of motor construction is presented in Figure 10f.

The last type of tested motor, called a pot-like microfluidic rotational vibrational motor with rigidly attached disc to the rotating axis, again used the principle presented in Figure 7. The central pillar was not attached to the glass support plate rigidly but had a small bearing for the rotating axis in the shape of a small hole drilled into the glass support plate. The other bearing was made as a hole in the upper plastic foil. Both bearings presented much less friction than the bearings presented in previous cases (see Figure 10e,f). The non-floating disc was attached rigidly with UV glue to the rotational axis directly below the surface of the water in the pot. Here, the disc was made from plastic foil with a density greater than the water density. The covering transparent plastic foil, was again used to prevent the evaporation of the liquid from the inside of the pot, and to allow the observation trough the foil into the pot to see the rotation of the disc. The cross-section of this type of motor construction is presented in Figure 10g.

#### 2.3.5. A Method for Measurement of the Amplitude and Phase Shift between Vibrations in Both *x* and *y* Directions

The method for measurement of amplitude and phase shift of circular vibrations was published in [17]. The method was based on the observation of the micro-sized circular-shaped object’s vibrational movements glued on the upper surface of the glass support plate situated nearby the pot. A polystyrene sphere with the diameter *d* = 30 µm was used as the object (see Figure 11a). The diameter *d* was measured when both piezo actuators were switched off. Next, distances *x* and *y* were measured from an elliptical shape (see Figure 11b), when both actuators were switched on. Both vibration amplitudes, the amplitude in the x- direction $a_{1}$_,_ and the amplitude in the direction of the longer diameter of the ellipse $a_{2}$, were calculated by Equation (13):(13)$$a_{1}=\frac{x-d}{2},{\;a}_{2}=\frac{y-d}{2}\;.$$

The phase shift *α* was measured between the *x* axis and the direction of the longer diameter of the ellipse (see Figure 11b). When the regular circular shape (*x* and *y* had the same length) was measured, instead of the elliptical shape, then the phase shift was 90° or −90°, depending on the direction of the water streaming in the pot. 

#### 2.3.6. A Method for Measurement of the Rotational Speed of the Water Streaming in the Pot

First, several rotations of the disc in the pot were filmed by the fast camera attached to the microscope. Then the film was reproduced in slow motion mode, so we were able to see the slow rotation of the filmed disc. The observation of frame by frame of the film’s consecutive images captured by the fast camera (1 frame/ms), gave us the time Δt needed for one revolution of the disc. At the end, the rotational speed was calculated with Equation (14):(14)$$v_{r}\left[\frac{rad}{s}\right]=\frac{2\cdot \pi }{\Delta t}.$$

The accuracy of the method was ± 1 time of the frame per revolution, that is 0.2%.

## 3. Results

We performed all seven experiments presented in Section 2.3.4.

### 3.1. Vibrational Pot-Like Mechanism

We started with the simplest experiment, where the water inside the pot was rotated with the changeable rotational speed around the pot axis in the z-direction (see Figure 10a). The center of rotation moved around the *z*-axis a bit by approximately 10 µm. The reason for that was probably not the constant vibrational amplitudes $a_{1}$ and $a_{2}$ and the angle *α* between both directions of amplitudes which varied due to disturbances from the environments (vibrations, temperature, changeable amplification ratio of power electronics due to temperature change, amplifier ratio also decreased with the rising frequency after approximately 500 Hz with 20 dB per decade …). 

It was also observed that the streaming of the water around the center was also a bit elliptical. The regular circle was, in fact, very rare. But the results of this experiment were very promising for the next performed experiments. The maximal rotational speeds in both directions were between −19 rad/s and +20 rad/s, measured at frequency *f* = 400 Hz, and the amplitudes of both actuators $a_{1}$ and $a_{2}$ at 8 µm and 10 µm, respectively. The rotational speed *v_r_* was variable and well controlled, with almost linear dependency between the voltage amplitude of the sine and cosine signals to the piezo actuators *V_pp_* and rotational speed of the streaming water *v_r_* (see Figure 12). The average values and Standard Deviation for every measurement point in Figure 12 were calculated from ten measurements. The linear dependency between the rotational speed of the water streaming in the pot and applied sine and cosine voltage *Vpp* to piezo actuators allowed us simple control of the pot-like vibrational micromotor. 

The direction of water streaming was simply changed with the change of phase between sine in cosine voltages from 270° to 90°, as was demonstrated in the Appendix A. 

We observed that the rotational speed of the water inside the pot was heavily dependent on the vibrations’ frequency in the real lab experiment, which was not predicted by the mathematical model described by Equations (1)–(12). The streaming of water at some frequencies stopped completely. Sometimes we also had two, or even more, small rotating whirls in the pot, especially at higher frequencies between 650 Hz and 1000 Hz. Therefore, we measured the amplitudes $a_{1}$ and $a_{2}$, phase shift angle *α* between both directions of amplitudes and rotational speed $v_{r}$ as a function of the applied vibration frequency *f*. The measured data are presented in Figure 13. The average values and Standard Deviation for every measurement point in Figure 13a–c) were calculated from ten measurements. The resonant peak of rotational speed of the whirling water in the pot between frequencies 300 Hz to 600 Hz (see Figure 13c) was measured once again, but every 20 Hz, again with 10 measurements at every measurement point. The average calculated rotational speeds $v_{r}$ and their Standard Deviation at every measurement point (every 20 Hz) are presented in Figure 14.

### 3.2. Free Floating Rotating Disc

The next experiment presented the ability of the whirling water in the pot to rotate the free-floating disc in the center of the pot (see Figure 10b). The disc remained in the center of the whirl more or less all the time during the experiment, due to the convex shape of the water’s surface. The free-floating disc rotated with the rotational speed $v_{r}\;$= 10 rad/s at frequency *f* = 380 Hz. Measured vibration amplitudes were: $a_{1\;}$= 7 µm and $a_{2}\;$= 9 µm. The disc, made from the Styrofoam, was floating on the top of the water surface with the outer edges of the disc submerged. The rotational speed of the disc was approximately 4–5 rad/s lower for the frequencies of vibrations between 300 Hz to 600 Hz than the measured velocities in the previous experiment (see Section 3.1, Figure 14). 

The experiment was demonstrated in Appendix A.

### 3.3. Pot-Like Microfluidic Rotational Vibrational Motor with Concave Shape of the Water’s Surface

The goal of the third experiment was to achieve transferring the rotational mechanical power out of the pot with the central mounted axis (see Figure 10c). The central axis (diameter approx. 80 µm) was fixed in the bearings made by the glass tube with inner diameter of 100 µm. The problem of this type of microfluidic motor was that it was not able to start to rotate by itself. It needed a little help, which was done by a gentle push by tweezers. The reason for that was too big static friction in the bearings. Later, when the disc started to rotate, the rotational speed achieved $v_{r}\;$= 25 rad/s at frequency *f* = 500 Hz. We tried to decrease the length of the glass tube (the bearings) and, consequently, decrease the static friction during the start of rotating. However, in this case, the axis wobbling was preventing the attachment, for example, the gear box on the axis.

Normally, the shape of the water surface in the pot with plastic walls is convex. The concave shape of the water surface was achieved simply by adding the liquid drop by drop until the concave shape was formed above the upper end of the pot. The speed of the water rotation velocity in the case of the concave shape of the water’s surface was a bit higher, by about 3–4 rad/s, than when the shape of the water surface was convex. The reason for that was that the concave shape of the water surface had lower friction with the wall of the pot. However, a disadvantage of the concave shape of the water’s surface was the accelerated evaporation of the water, which practically limited the existence of the concave shape of the water surface and, therefore, faster rotation to about 15–20 min, only. 

The experiment is presented in Appendix A.

### 3.4. Pot-Like Microfluidic Rotational Vibrational Motor with Centrally Mounted Floating Disc on the Central Pillar with Radius 40 µm

We tried to develop the microfluidic motor without the wobbling of the disc, or the axis attached to the disc in the fourth experiment—the pot-like microfluidic vibrational motor with centrally mounted floating disc on the central pillar (see Figure 10d). The solution with the central pillar in the pot had two advantages against the previous three solutions described in Section 3.1, Section 3.2, Section 3.3.

The first one fixed the disc in the center of the pot, while the other ones increased the rotational speed of the whirling water in the pot. The construction solution with the central pillar in the middle of the pot increased the rotational speed of the whirling liquid in the pot, as was established in the simulation investigation (see Section 2.3.3, Figure 7 and Figure 8). The floating disc was rotating with the rotational speed $v_{r}\;$= 30 rad/s at frequency *f* = 500 Hz and amplitudes of vibrations $a_{1}\;$= 13 µm and $a_{2}\;$= 9 µm. The maximum rotational speed in the clockwise direction (CW) was approx. 30 rad/s, while the maximum speed in the counterclockwise direction (CCW) was approx. -29 rad/s. It started to move without problems, so we believed that this experiment could also be the basis for the next two types of microfluidic motors. The experiment was presented in Appendix A. 

Unfortunately, the next two experiments (centrally mounted floating disc on the central pillar with radius 40 µm and attached outer rotating axis (see Figure 10e), and centrally mounted floating disc on the central pillar with radius 120 µm and attached outer rotating axis (see Figure 10f)), were not successful. Both microfluidic motors did not start to rotate, even if we tried to help them to start by pushing the disc with tweezers. Obviously, the friction in the bearings was too big, but we learned a lot from these two unsuccessful experiments, especially the need to make the bearings with lower friction.

### 3.5. Pot-Like Microfluidic Rotational Vibrational Motor with Rigidly Attached Disc to the Rotating Axis

We learned from the previous two unsuccessful experiments with the construction solutions presented in Figure 10e,f, and explained at the end of the Section 3.4, how to reduce the friction in the bearings. The friction between the axis and outer layer of bearings could be reduced if the contact surface between the axis and the outer layer of the bearings was made as small as possible. So, we drilled a small pit in the glass supported plate, which was used as the centering hole of the lower bearings. The hole through the upper transparent plastic foil was used as the outer part of the upper bearings. The contact surfaces between both outer surfaces and the centrally mounted axis were made as small as possible and, consequently, the static and viscous frictions in the bearings were reduced. Such a construction of the bearings allowed us to start rotation and rotating the disk in the pot without problems. The construction of this type of microfluidic motor—a pot-like microfluidic rotational vibrational motor with the disc attached rigidly to the rotating axis is presented in Figure 10g. Here the innovation was made with the upper plastic foil, which covered and sealed the top of the pot completely. This innovation almost prevented the evaporation of the water from the pot. The evaporation was still possible through the upper bearings, but it was much lower than in the case when the upper plastic foil was not sealing the pot. For example, a half of the water in the pot evaporated in an hour when the plastic foil did not seal the pot, while the same quantity of the water evaporated from the sealed pot in 6 days. Such a construction also almost prevented the axis from wobbling. The disc was not floating on the surface anymore, but was submerged under the surface, so we were not limited anymore to producing the disc from the Styrofoam material. The disc was made from the plastic foil, which had a density slightly greater than water. The disc was rotated with the rotational speed $v_{r\;}$= 14 rad/s at vibration amplitudes $a_{1}\;$= 4 µm, $a_{2}\;$= 7 µm, shift angle *α* = 85° and frequency *f* = 420 Hz. The maximum rotational speed in the CW direction was approx. 14 rad/s, while the maximum rotational speed in the CCW direction was approx. −15 rad/s. The experiment is presented in Appendix A.

Surprisingly, the measurements of rotational velocity $v_{r}$ resonant peak versus frequency in the pot (see Figure 14) changed a lot for this last type of microfluidic motor. In fact, we had two resonant peaks in the frequency range from 200 Hz to 500 Hz. This was probably a consequence of the changed construction of the mechanism of this type of microfluidic motor in comparison with the motor described in Section 3.1, Section 3.2, and Section 3.3. The change of vibration amplitudes $a_{1}$, $a_{2}$_,_ the phase shift angle *α* and rotational velocity of the disc inside the pot versus the frequency of vibrations in the frequency region from 200 Hz to 500 Hz are presented in Figure 15.

Two microfluidic rotational motors of this type (see Figure 10g) were built with almost the same dimensions, because the glass supported plate for the first one was broken accidentally. The second one worked as perfectly as the first one, but the position and width of the resonant peaks in the frequency region 200 Hz to 500 Hz were changed in comparison to those presented in Figure 15c. Probably the slightly difference in the shape of glass supported plate, and, therefore, the weight of it, caused the difference. 

## 4. Discussion

### 4.1. Understanding the Changes of the Resonant Peak

The mathematical model described with Equations (1)–(12) predicted that the rotational speed would be increased by increased vibrational frequency *f* with a cubic relationship, and by the increased amplitude of vibrational frequency *A* with a quadratic relationship (see Equation (2)). So, according to the mathematical model, the highest the amplitude and frequency were, a higher rotational speed could be achieved of the whirling water. Of course, the mathematical model did not contain the model of the electronic amplifier and the mechanical part of the motor. In reality, the frequency and phase characteristics of the amplifier and the mechanical part produced the so-called resonant peak response, measured in Figure 13, Figure 14 and Figure 15.

The most challenging was to understand what kind of resonant peaks (see Figure 13, Figure 14 and Figure 15) produced the most suitable whirling of the water during the measurements of vibration amplitudes $a_{1}$,$\;a_{2}$, the phase shift angle *α* and rotational velocity of the disc inside the pot versus the frequency of vibrations. After experimentation with different sizes (diameter, height) of the pot or disc, or the thickness of the supporting glass plate, it was discovered that the shape of the resonant peaks changed a lot. The resonant peaks of vibration amplitudes $a_{1}$, $a_{2}$, the phase shift angle *α* and rotational velocity of the disc inside the pot versus the frequency of vibrations for two applications presented by Figure 10c,g were different according to the shape and the frequency of the resonant peak or peaks (see Figure 13, Figure 14 and Figure 15). Nevertheless, they had the same main characteristics: The vibration amplitudes $a_{1}$ and $a_{2}$ must both be as much higher than zero, and the phase shift angle α must be as close to 90° as possible at the frequency nearby the resonant peak. This kind of combination of the vibration amplitudes and phase shift angle produced the stable and high-speed circular whirling of the water in the pot. If the phase shift angle was inside the regions between 80°–85° and 95°–100°, then the circular shape of the whirls changed to an elliptic shape. If the shift angle α was lower than 80° or higher than 100°, then the whirling of the water in the pot stopped completely, or sometimes even produced two, or several whirls inside the pot, in spite of both vibrational amplitudes $a_{1}$ and $a_{2}$ being equal and high. A similar effect was discovered if the vibration amplitudes $a_{1}$, $a_{2}$, were unequal with the shift angle close to 90°. The elliptic shapes of the whirls produced lower rotational speed of the disc and, consequently, reduced the maximal torque of the motor.

We discovered during the experimentations that the shape of the resonant peaks and their amplitudes could be changed partly, or even improved, with changing of the mass of the disc, height of the pot, quantity of water in the pot and mass of the glass plate. Sometimes we even succeeded in moving the resonant peaks of the amplitudes and angles to higher frequencies, and, consequently, increased the rotational speed of the water by simply adding or moving some small weights on the supporting glass plate. All these experiments have proven that the resonant peak frequency is also dependent on the mechanical properties of the pot-like mechanism, and not only of the properties of the electronic amplifier alone.

### 4.2. Estimation of Motor Maximum Torque of the Two Types of Microfluidic Motor

Estimation of the maximum torque for two types of microfluidic motors, presented by Figure 10d,g, was performed with the method published in [3]. The method measured the time which was needed to change the rotational speed from maximal speed in a CW direction to maximal speed in a CCW direction for an unloaded and loaded microfluidic motor. The inertia of the unloaded and loaded rotor disc was calculated from the geometrical (outer and inner diameters of the disc and axes, heights of the discs and axes—see Figure 16) and mass properties (densities of the disc materials) of the rotor.

The maximum torque was calculated with Equation (15):(15)$$T_{\max}=T_{L}-T_{\mathrm{UL}}=J_{L}\cdot \frac{{\Delta \omega }_{L}}{{\Delta T}_{L}}+{\;\omega }_{L}\cdot \;B+F-{\;J}_{\mathrm{UL}}\cdot \frac{{\Delta \omega }_{\mathrm{UL}}}{{\Delta T}_{\mathrm{UL}}}-{\;\omega }_{\mathrm{UL}}\cdot \;B-F$$
where $T_{L}$ and $T_{UL}$ were loaded and unloaded rotor torques. $J_{L}$ and $J_{UL}$ presented the inertia of both the loaded and unloaded inertia rotors, *B* was the viscous friction in the bearings, *F* presented the Coulomb friction in the bearings, while $\Delta \omega_{L}$ and $\Delta \omega_{UL}$ were changes between maximal rotational speeds in both CW and CCW directions for the loaded and unloaded rotor. $\Delta t_{L}$ and $\Delta t_{UL}$ presented the time needed to change from maximal rotational speed from the CW to maximal rotational speed in the CCW direction for the loaded and unloaded rotor, and $\omega_{L}$ and $\omega_{UL}$ were maximal speeds for both loaded and unloaded rotors, respectively. 

We performed the measurements, where loaded rotor inertia was increased steadily. The measured time $\Delta T_{L}$ increased, consequently, while maximal speed $\omega_{L}$ remained unchanged. We chose the maximal loaded rotor inertia $J_{L}$ when the speed was still unchanged. If we still increased the loaded rotor inertia $J_{L}$ then the maximal speed started to decrease. Therefore, we can make $\Delta \omega_{L}\;=\;\Delta \omega_{UL}=\Delta \omega$ and $\omega_{L}\;=\;\omega_{UL}$, and therefore Equation (15) could be simplified into Equation (16):(16)$$T_{\max}=J_{L}\cdot \frac{\Delta \omega }{{\Delta T}_{L}}-{\;J}_{\mathrm{UL}}\cdot \frac{\Delta \omega }{{\Delta T}_{\mathrm{UL}}}.$$

Case 1: Introducing inertia $J_{L}$ and $J_{UL}$ for the shape of the rotor presented in Figure 16a,b for the microfluidic motor with floating disc (see Figure 10d) gave the Equation (17): (17)$$T_{\max}=\frac{\pi }{2}\left\{\left[\rho_{\mathrm{Sf}}h_{\mathrm{Sf}}\left(r_{2\mathrm{Sf}}^{4}-r_{1\mathrm{Sf}}^{4}\right)+\rho_{\mathrm{Al}}h_{\mathrm{Al}}\left(r_{2\mathrm{Al}}^{4}-r_{1\mathrm{Al}}^{4}\right)\right]\frac{\Delta \omega }{{\Delta t}_{L}}-\rho_{\mathrm{Sf}}h_{\mathrm{Sf}}\left(r_{2\mathrm{Sf}}^{4}-r_{1\mathrm{Sf}}^{4}\right)\frac{\Delta \omega }{{\Delta t}_{\mathrm{UL}}}\right\}=0.222\;\mathrm{pNm},\;$$
where $h_{Sf}\;=\;$200 µm, $h_{Al}\;=\;$140 µm, $r_{2Sf}\;=\;$175 µm, $r_{1Sf}\;=\;$50 µm, $r_{2Al}\;=$ 195 µm, $r_{1Al}\;=\;$70 µm, $\Delta \omega \;=\;59\frac{\mathrm{rad}}{s}$ (see Section 3.4), $\Delta t_{L}\;=\;$0.22 s, $\Delta t_{UL}\;=\;$0.11 s, while $\rho_{Sf}\;=\;$50 kg/m^3^ and $\rho_{Al\;}\;=\;$2700 kg/m^3^ presented the density of Styrofoam and aluminum, respectively.

Case 2: Introducing inertia $J_{L}$ and $J_{UL}$ for the shape of the rotor presented in Figure 16c,d for the microfluidic motor with submerged disc fixed on the glass axis (see Figure 10g) gave the Equation (18): (18)$$T_{\max}=\frac{\pi }{2}\left\{\left[\rho_{\mathrm{Pl}}h_{\mathrm{Pl}}\left(r_{2\mathrm{Pl}}^{4}-r_{1\mathrm{Pl}}^{4}\right)+\rho_{\mathrm{Al}}h_{\mathrm{Al}}\left(r_{2\mathrm{Al}}^{4}-r_{1\mathrm{Al}}^{4}\right)\;+\rho_{\mathrm{Gl}}h_{\mathrm{Gl}}r_{\mathrm{Gl}}^{4}\right]-\;[\rho_{\mathrm{Pl}}h_{\mathrm{Pl}}\left(r_{2\mathrm{Pl}}^{4}-r_{1\mathrm{Pl}}^{4}\right)+\rho_{\mathrm{Gl}}h_{\mathrm{Gl}}r_{\mathrm{Gl}}^{4}]\frac{\Delta \omega }{{\Delta T}_{\mathrm{SUL}}}\right\}=0.101\;\mathrm{pNm},\;$$
where $h_{Pl\;}=\;$100 µm, $h_{Al}\;=\;$185 µm, $h_{Gl}=\;$810 µm $r_{2Pl}\;=\;$165 µm, $r_{1Pl}\;=\;$50 µm, $r_{2Al}=$195 µm, $r_{1Al}\;=\;$50 µm, $r_{Gl\;}=\;$80 µm, Δω = 29 rad/s (see Section 3.4), $\Delta t_{L}\;=\;$0.23 s, $\Delta t_{UL}=\;$0.10 s, while $\rho_{Pl}\;=\;$1050 kg/m^3^, $\rho_{Gl}\;=\;$2500 kg/m^3^ and $\rho_{Al\;}\;=\;$2700 kg/m^3^ presented the density of plastic, glass and aluminum, respectively.

The estimations of the maximal torque in both previous cases were done under the assumption that both loaded and unloaded rotor rotational speeds of both motors in Cases 1 and 2 reached the maximal (equal) rotational speeds in both the CCW and CW directions. Of course, the motors worked well if the rotational speeds of the loaded rotors were also lower than the maximal rotational speeds of the unloaded rotors. In this case, the maximal torque was higher than those estimated by Equations (17) and (18), but with lower dynamics. The rotor rotational speed measurements were done with the use of a highspeed camera attached to the microscope. The measurements of the rotational speed (Standard Deviation was approx. 20%) and, therefore, measurements of *ΔT_Sf_* and *ΔT_SfAl_* ((Standard Deviation was approx. 25%)) were quite problematic. This was also the reason why the estimation of maximal torque of the motors in Cases 1 and 2 were not particularly accurate, due to substantial measurement errors, but they were in the range of from 0.1 pN to 0.2 pNm. The Matlab code of the load inertia and *T_max_* calculations are attached as Appendix A.

### 4.3. Stability of Rotational Speed

If the pot (the outer envelope of the motor) was open, the water evaporated from the height of 1000 µm to the height of approximately 500 µm in an hour of the motor running. During the decreasing of the height of water in the pot the maximal rotational speed also decreased. This effect was especially obvious when the height of the water was lower than approx. 700 µm. This was the reason why we wanted to close the pot from above. We made several experiments to check the rotational speed stability at fixed frequency of the experiment. Amplitudes of vibrations $a_{1}$, $a_{2}$_,_ and also shift angle *α*, started to decrease with the lower height of water in the pot and, consequently, the speed also decreased. However, the most devastating disturbance occurred when the height of the water was lower than 500 µm, when several vortices appeared in the pot and stopped the rotating of the disc completely. This was probably the influence of the vibrating glass supporting plate. If the water height in the pot was increased above 1 mm up to 1.5 mm, then the speed remained unchanged. The change of temperature also produced a minor change of the rotational speed because the kinematic viscosity η of the water is highly dependent on the temperature (see Section 2.3.1).

If the pot of the microfluidic motor was closed (sealed), then the water evaporation stopped completely, and the rotation speed of the rotor remained unchanged for days, or weeks if necessary. This experiment was performed with the microfluidic motor with the floating disc (see Figure 10d) in the sealed pot. Of course, in the case of the microfluidic motor with the submerged disc attached rigidly to the axis, which was used to transfer the mechanical rotational energy out of the pot, the sealed pot was not an option. The small hole used as an upper bearing (see Figure 10g) enabled the slowed down evaporation. The height of the water decreased from 1 mm down to 0.7 mm in the pot in about 8-9 h at the temperature of the surrounding air (cca. 22 °C) if the non-stop experiment was performed during the mentioned time. 

### 4.4. Possible Improvements

We performed several experiments with rotor diameters greater than 350 µm and diameters of the pots greater than 500 µm. This was a way to increase the maximal torque but, unfortunately, the maximal rotational speed was decreased. For example, if the diameter of the pot was 2 mm with the rotor diameter 1 mm, then only a rotational speed of the rotor of 3–4 rad/s was achieved, and that was only for the application with the floating disc. In the case of the submerged disc fixed on the axis, the application did not work at all. Obviously, the friction force in the bearings was too high. In fact, the highest rotational speed was achieved with the rotor diameters between 300 and 400 µm and the pot diameters between 500 to 600 µm. In these cases, the friction force in the bearings did not present an insurmountable obstacle for the three experimentally presented microfluidic motor applications (presented with Figure 10c,d,g).

The following possible improvements should be done to solve the two main problems:Prevent the evaporation of the liquid from the pot: using low evaporation rate solvents (Cyclohexanol, Eastman 2-ethylhexanol, …), and/or make better (more sealed) upper bearings. We experimented by putting a small drop of lubricant on the place of the upper bearings’ hole through the upper plastic foil in the case of the microfluidic motor with the submerged disc fixed on the axis (see Figure 10g). In this case, we achieved a slightly higher rotational speed at the beginning of the experiment, due to reducing the friction in the upper bearings, and also lowered the rate of evaporation. We prolonged the non-stop operation of the microfluidic motor from about 9 h, let us say to about 10 h. We would probably need more viscous lubricant to prolong the working time. Our type of lubricant somehow penetrated through the upper bearings, due to vibrations, and polluted the water in the pot. Due to the increased kinematic viscosity of the polluted water, the maximal rotational speed was reduced after 4–5 h of working time by about 10–15%.More quality bearings with smaller and a smoother hole in the bearings and lubricant in the bearings, should be used to prevent the axis wobbling. 

Of course, the rotational speed and maximal torque should be increased by working with a better electronic amplifier. Our amplifier (presented in Section 2.2) had the cut off frequency approximately at 500 Hz, so higher frequencies did not produce enough vibration amplitudes $a_{1}$ and *a*_2_. A better electronic amplifier would also prevent the shift angle *α* from being remarkably reduced at a higher frequency than 500 Hz (see Figure 13, Figure 14, and Figure 15). 

We observed the overheating of the piezoelectric actuators, especially if non-stop running of the microfluidic motor was performed for longer than one hour. So, it would be necessary to construct a cooling system in the shape of cooling fins or a fan for the piezoelectric actuators for longer periods of running. 

### 4.5. Endurance Test

We were investigating the endurance of the microfluidic motor with the submerged disc fixed on the axis. We performed non-stop running of the microfluidic motor in both directions, five minutes in a CCW direction, and another five minutes in a CW direction, with maximum rotational speed for 8 h, for three days consecutively. Every hour we stopped the running of the motor only to cool down the temperature of the piezoelectric actuator for 15 min. After one hour, the piezoelectric actuator reached the temperature around 150 °C. During the first two days, the system was working without problems. On the third day the test lasted only 7.5 h. We had one failure because the glued connections between the glass supporting plate and the tip of the second piezoelectric actuator was broken. After an inspection of the complete device, we concluded that there were no particular problems with the microfluidic motor’s subsystems, except for the overheating of the piezoelectric actuators. No wear was seen in the lower and upper bearings during the inspection. 

## Figures and Tables

**Figure 1 micromachines-12-00177-f001:**
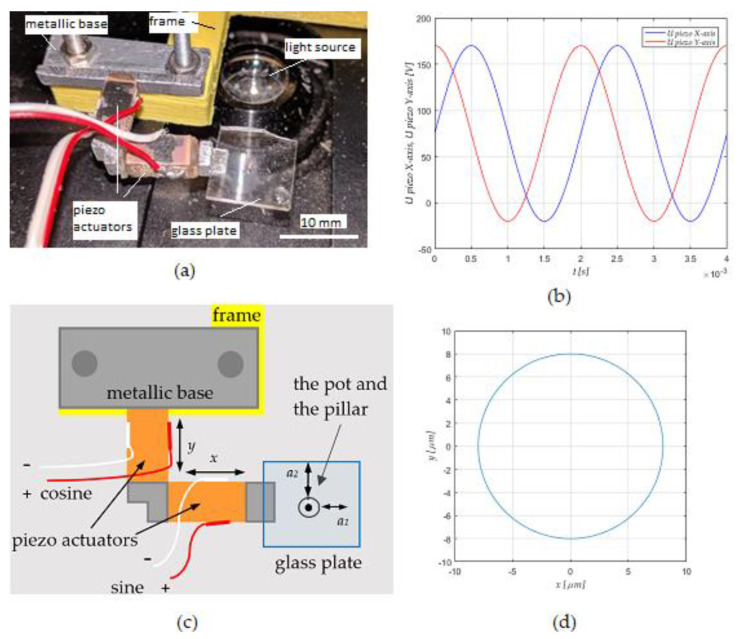
(**a**) Mechanical part of the lab set-up with two perpendicularly attached piezo actuators, (**b**) cosine and sine sinusoidal voltage supply voltage *V_pp_* for the piezo actuators, (**c**) the scheme of the mechanical part, and (**d**) circular vibrational movement.

**Figure 2 micromachines-12-00177-f002:**
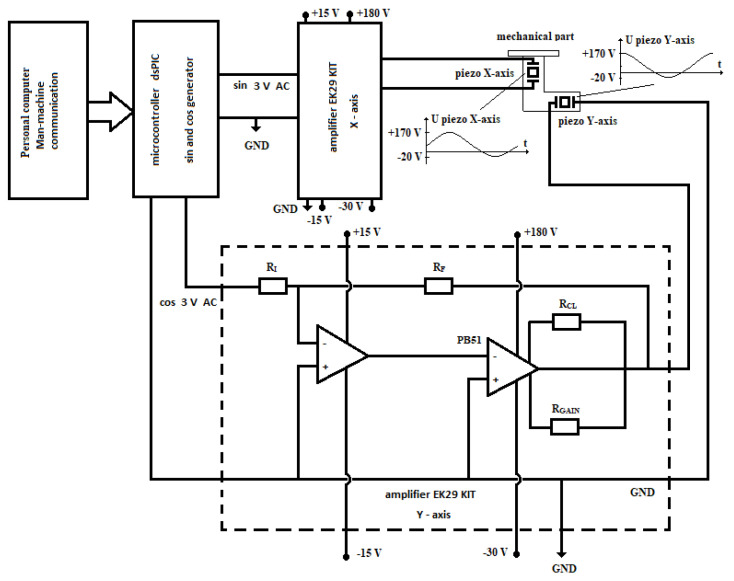
Electronic part of the laboratory set-up with two EK29 KIT high voltage amplifiers.

**Figure 3 micromachines-12-00177-f003:**
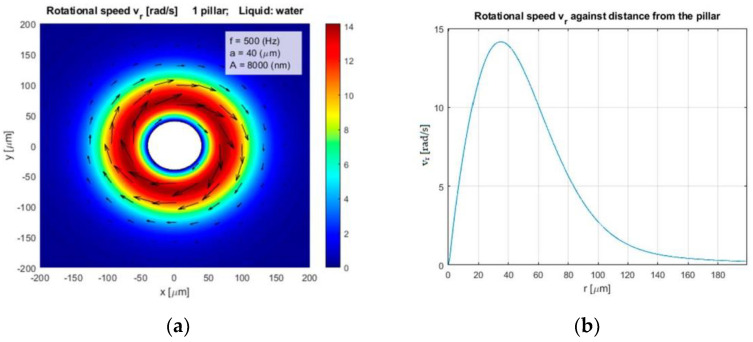
(**a**) Rotational velocity of water stream $v_{r}$ and, (**b**) its amplitude as a function of distance from the pillar.

**Figure 4 micromachines-12-00177-f004:**
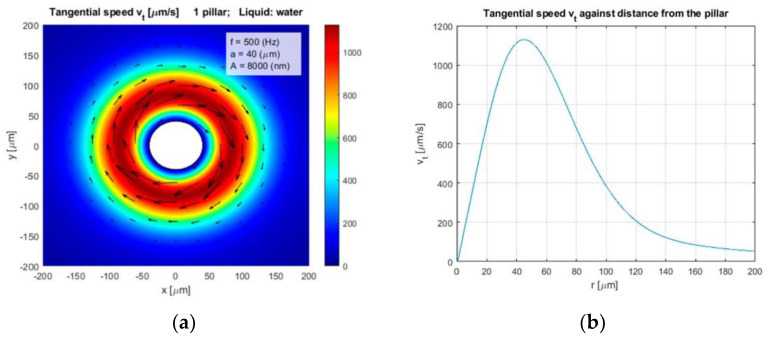
(**a**) Tangential velocity of water stream $v_{t},$ and (**b**) its amplitude as a function of distance from the pillar.

**Figure 5 micromachines-12-00177-f005:**
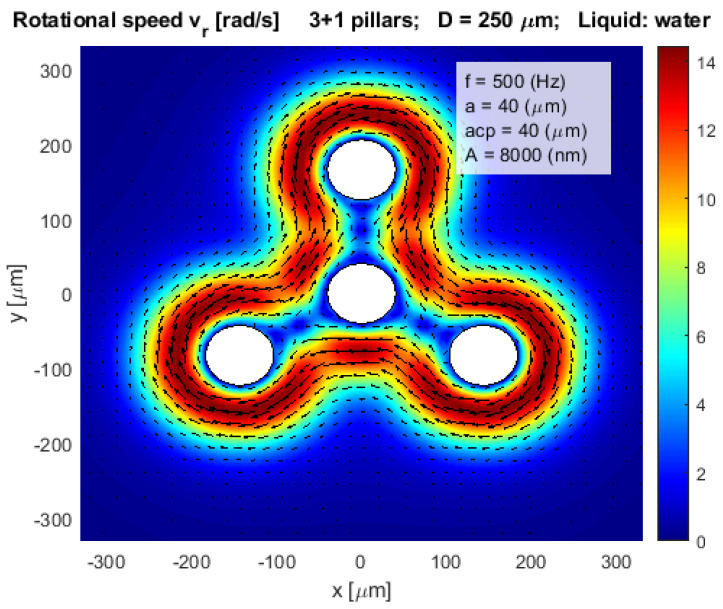
Rotational velocity of water stream $v_{r}$ in a more complex layout.

**Figure 6 micromachines-12-00177-f006:**
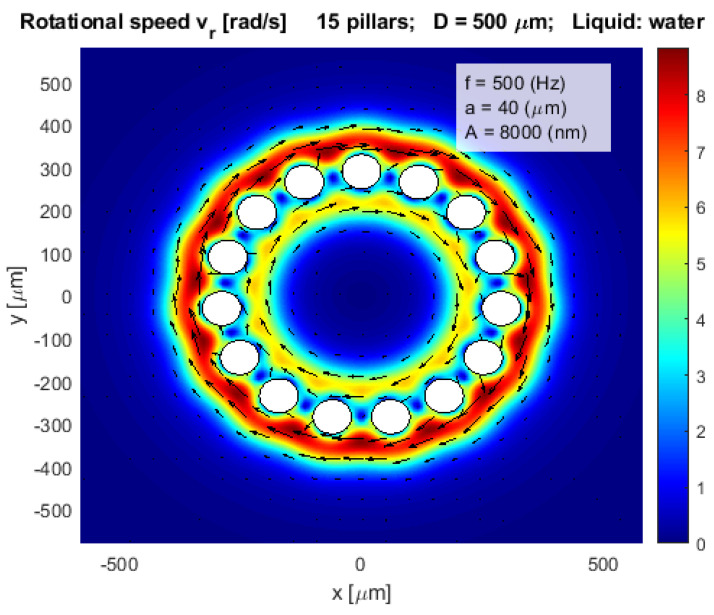
Rotational velocity of water stream $v_{r}$ in an open pot-like layout.

**Figure 7 micromachines-12-00177-f007:**
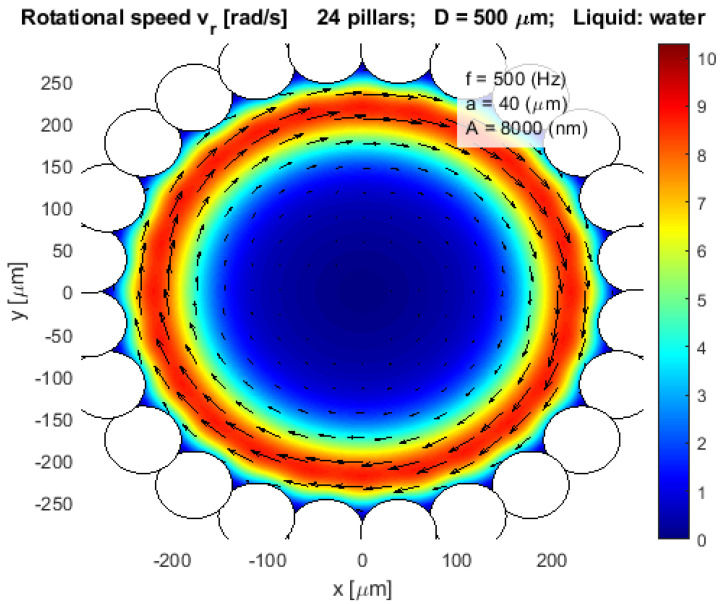
Rotational velocity of water stream $v_{r}$ in a closed pot-like layout.

**Figure 8 micromachines-12-00177-f008:**
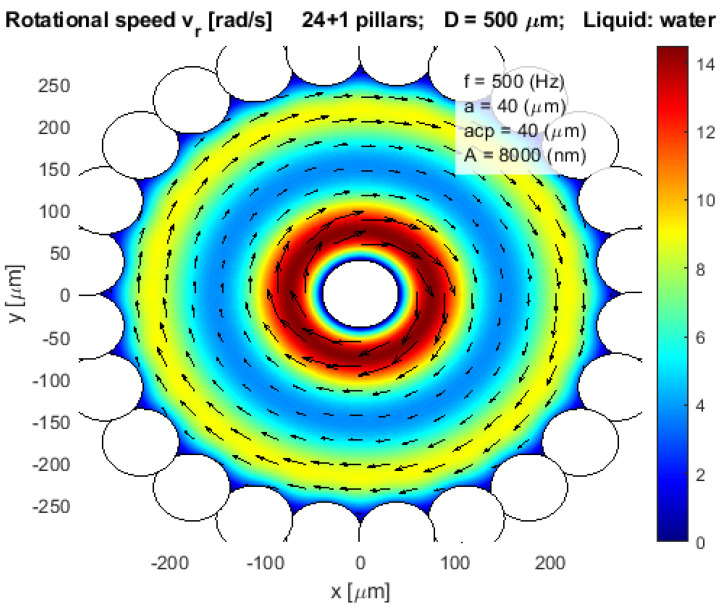
Rotational velocity of water stream $v_{r}$ in a closed pot-like layout with a central pillar.

**Figure 9 micromachines-12-00177-f009:**
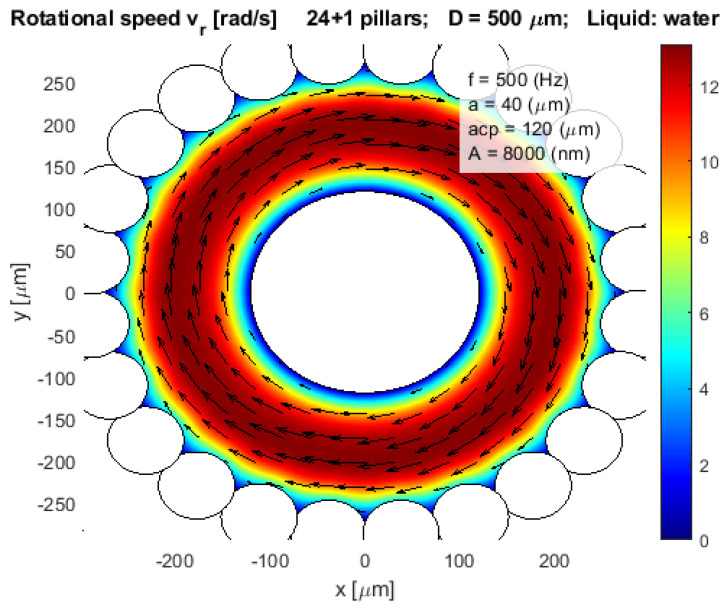
Rotational velocity of water stream $v_{r}$ in a closed pot-like layout with a large central pillar.

**Figure 10 micromachines-12-00177-f010:**
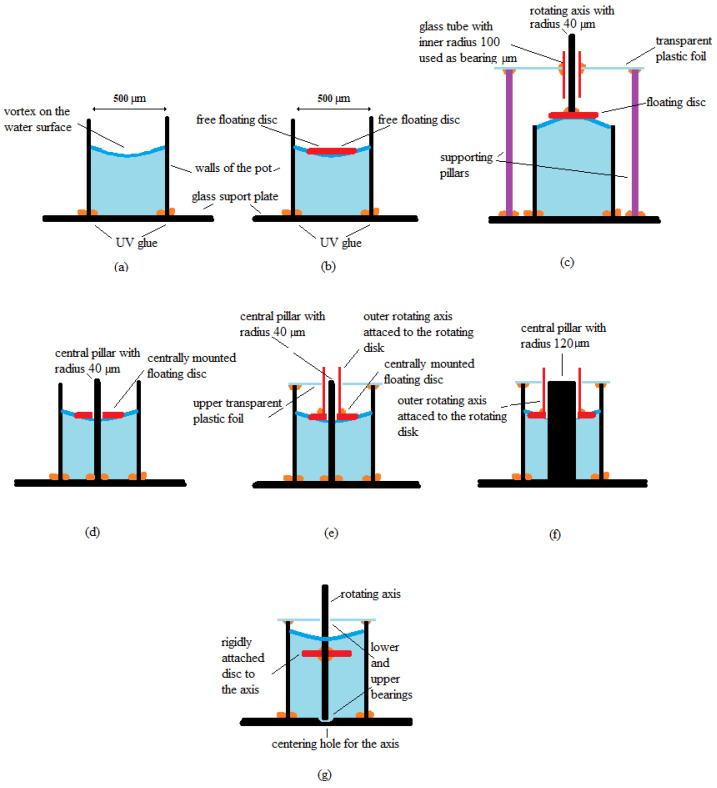
Seven tested types of pot-like microfluidic rotational vibrational motors with: (**a**) Vibrational pot-like mechanism. (**b**) Free floating rotating disc. (**c**) Pot-like microfluidic rotational vibrational motor with concave shape of the water surface. (**d**) Centrally mounted floating disc on the central pillar with radius 40 µm. (**e**) Centrally mounted floating disc on the central pillar with radius 40 µm and attached outer rotating axis. (**f**) Centrally mounted floating disc on the central pillar with radius 120 µm and attached outer rotating axis. (**g**) Disc attached rigidly to the rotating axis.

**Figure 11 micromachines-12-00177-f011:**
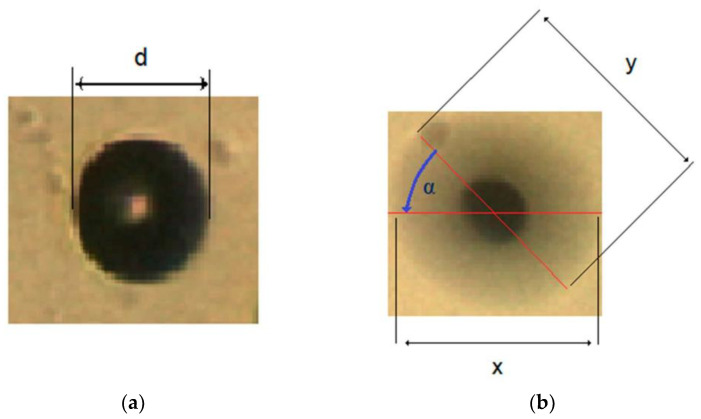
Measurements of vibration amplitudes and phase shift between directions of vibration amplitude: (**a**) Piezo actuators are switched off. (**b**) Piezo actuators in both x and y directions are switched on.

**Figure 12 micromachines-12-00177-f012:**
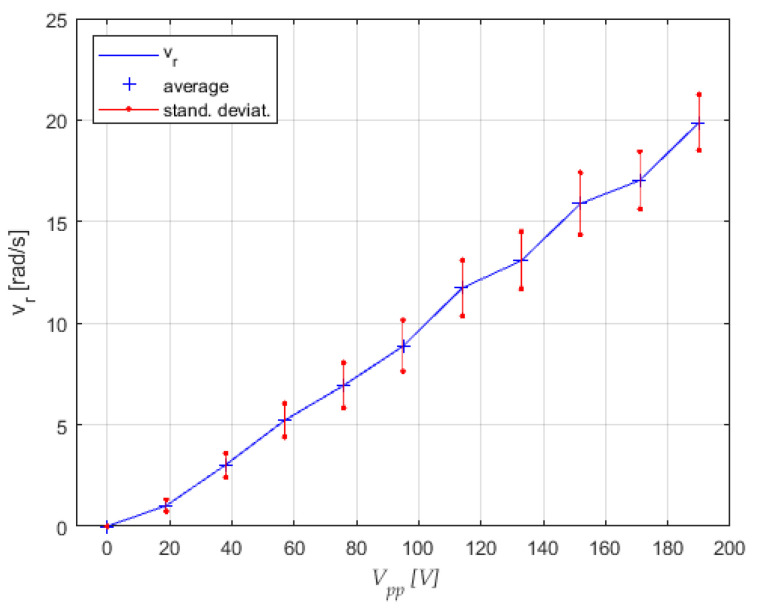
Measurements of the rotational speed of water streaming *v_r_* versus sine and cosine voltage amplitudes *V_pp_* at *f* = 400 Hz for the vibrational pot-like mechanism.

**Figure 13 micromachines-12-00177-f013:**
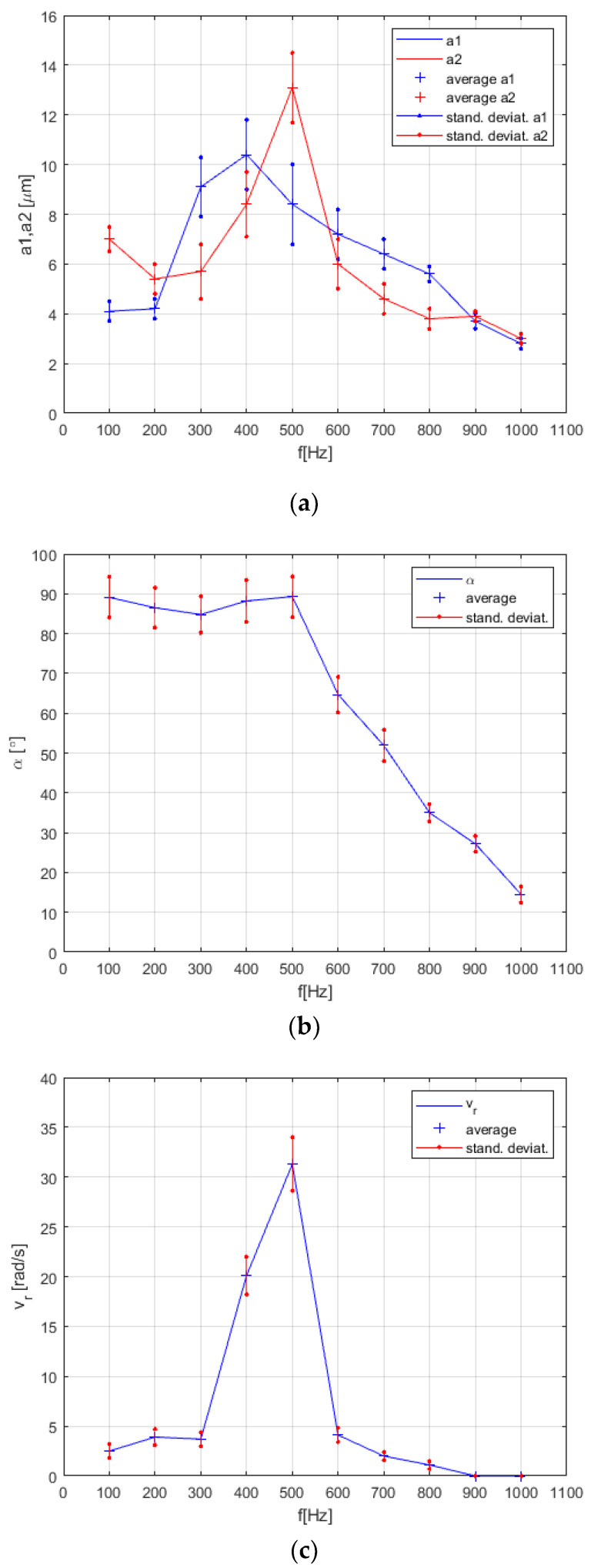
Measurements for the vibrational pot-like mechanism: (**a**) Vibration amplitudes $a_{1}$ and $a_{2}$ versus frequency *f*. (**b**) Phase shift α versus frequency *f*. (**c**) Rotational velocity $v_{r}$ versus frequency *f*.

**Figure 14 micromachines-12-00177-f014:**
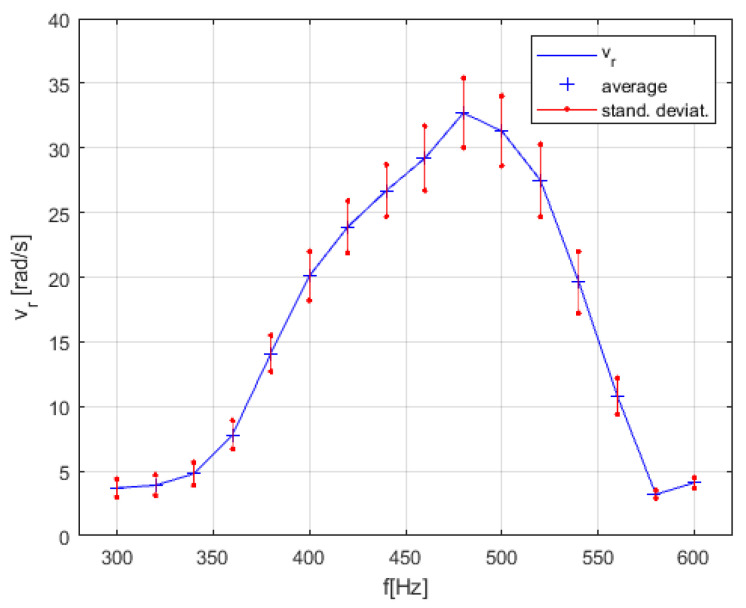
Accurate measurements of rotational velocity $v_{r}$ resonant peak versus frequency *f* for the vibrational pot-like mechanism.

**Figure 15 micromachines-12-00177-f015:**
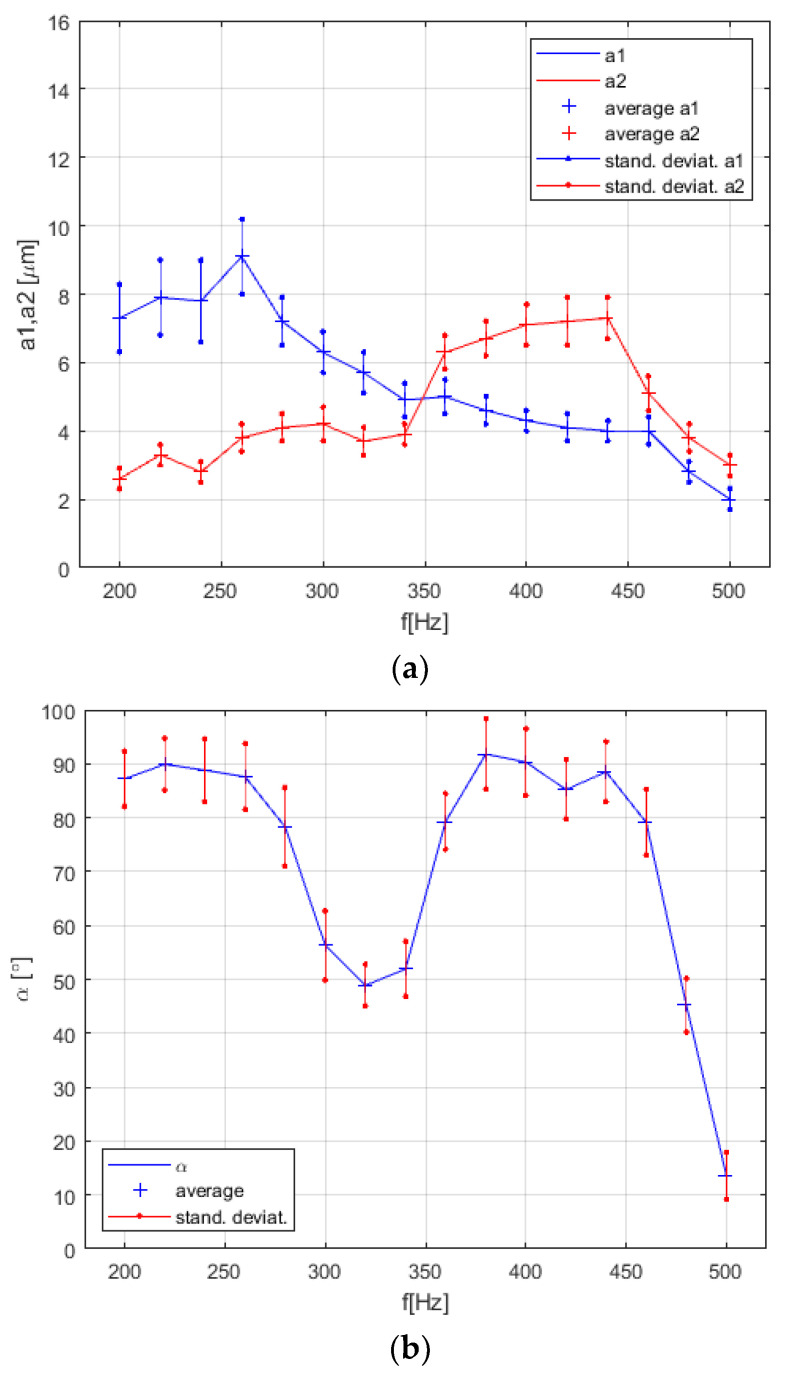
Measurements for the pot-like microfluidic rotational vibrational motor with rigidly attached disc to the rotating axis. (**a**) Vibration amplitudes $a_{1}$ and $a_{2}$ versus frequency *f*. (**b**) Phase shift *α* versus frequency *f*. (**c**) Rotational velocity $v_{r}$ versus frequency *f*.

**Figure 16 micromachines-12-00177-f016:**
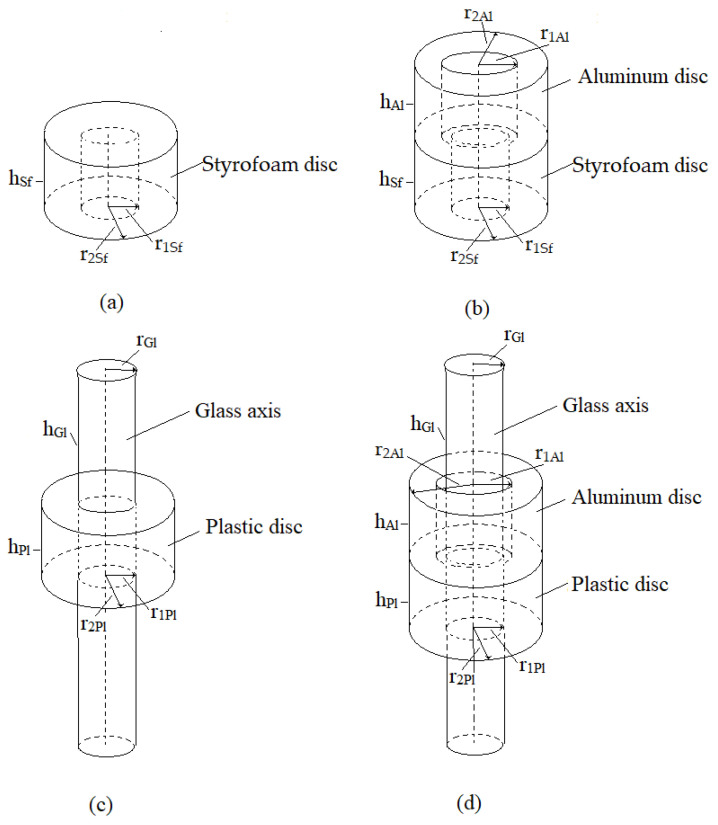
The schemes of the composite rotors. (**a**) Floating unloaded rotor. (**b**) Floating loaded rotor. (**c**) Submerged unloaded rotor. (**d**) Submerged loaded rotor.

**Table 1 micromachines-12-00177-t001:** Overview of microfluidic motors.

Ref.	Type	Rotor Size Diameter [µm]	Speed [rad/s]	Torque [pNm]	Controllability in Both Directions	Transfer of Mechanical Energy Out of the Motor
[1]	Microfluidic driven motor in the channel	60–1600	390	8.7	No	No
[2]	Electro-wetting micromotor	2000	18	-	Yes	No
[3]	Rotating the thin layer of liquid	-	9	-	Yes	No
[4,5,6]	Surface acoustic waves motor	5000	235	6 × 10^4^	Yes	No
[7]	Microfluidic motor with edges	600	125	-	No	Yes
[9]	Rotational Janus micromachine	100	0.6	-	No	No
[10,11]	Bubble-Powered Micro-Rotor	65–100	65–70		No	No
[17]	Microfluidic motor with central pillar	350	26	0.2	Yes	No
	**Our pot-like** **microfluidic motor**	**350**	**15**	**0.1**	**Yes**	**Yes**

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
