# Peer review of "A Pot-Like Vibrational Microfluidic Rotational Motor"

_micromachines, 2021, doi:10.3390/mi12020177_

Round 1

Reviewer 1 Report

A pot-like vibrational microfluidic rotational motor by Suzana Uran, Matjaž Malok, Božidar Bratina, and Riko Šafarič

In this paper, the authors used a mathematical model to guide and optimize the parameters and designs to build pot-like microfluidic rotational motors. The results are well organized and properly discussed. The bibliography is appropriate. Therefore, I recommend it for publication in Micromachines after some minor changes.

(1) Generally, we use the authors' names directly when citing or highlighting other researchers' work. For example, in line 41, "The authors in [3]" can be replaced with "Amjadi et al."

(2) The mathematical symbols are sometimes italic. I would keep them all italic. For example, in line 412, the frequency f is italic in the figure caption but it is not italic in the main text.

(3) Line 436, the abbreviations of "CW" and "CCW" should be made clear as "clockwise" and "counterclockwise".

Author Response

Please find the attached file with answers and added manuscipt with color changes in the text.

Reviewer 2 Report

A pot-like vibrational microfluidic rotational motor

Broader Comment: The paper presents an interesting rotational motor design and detailed experimentation. However, the authors have provided a lot of information with lengthy explanations, some of which might be redundant. The authors should update the manuscript to make it more concise. Additionally, the paper is presented like a technical report rather than a scientific article. Authors should focus on explaining the principles behind the technique and provide interpretation of their results. Individual results or values like amplitude, radial velocity achieved might vary based on the diameters of pillars or configuration, however if readers would like to replicate such a system, understanding the interpretation of the results will be more important than individual values. Please address the following concerns:

Specific Comments:

  1. Please provide a schematic to explain the setup for the materials section. Figure 1 shows the set up however, it is not clear how the actual device is operated.
  2. Provide background and explanation for equations 2-9
  3. How did you choose the pillar diameter for simulations? How is the amplitude affected by presence of multiple pillars in the configuration?
  4. In the circular pillar configuration, how was the number of pillars fixed? Was this done by trial and error? How did you select the amplitude and frequency for this configuration?
  5. It is not clear how the configuration in figures 7 and 8 with overlapping pillars can operate in an experiment. Is there no hindrance in motion due to the overlap?
  6. Please explain how the increase in overlap could cause the decrease in water streaming velocity (line 258-259). How did you obtain the optimum overlapping?
  7. Please explain Figure 12 and what does the trend of the plot represent. What do the two axes represent? How were these velocities measured?
  8. Lines 378-383: Please explain this clearly. Does this relate to what was observed in the simulations and from the mathematical model?
  9. Please explain how the convex/concave surface of water was obtained. How is that affecting your results?
  10. Please refine sections 3.4and 3.5. The explanation is not clear.
  11. Please provide interpretation of the Figures 13-15. What are these figures trying to convey to the reader? Why are they important to understand the system?

Author Response

(The authors gave the same response as above.)

Round 2

Reviewer 2 Report

The authors have answered questions and have updated the manuscript satisfactorily. Please proof read the manuscript to eliminate grammatical errors.